# Coming together in a digital age: Community twitter responses in the wake of a campus shooting

**Jessamyn Bowling**[1]*, **Erika Montanaro**[1], **Sarai Guerrero Ordonez**[1], **Sean McCabe**[1], **Shayna Farris**[1], **Neielle Saint-Cyr**[1], **Wade Glaser**[2], **Robert J. Cramer**[1], **Jennifer Langhinrichsen-Rohling**[1], **Annelise Mennicke**[1]

1 University of North Carolina at Charlotte, Charlotte, NC, United States of America, 2 Gaston Day School, Gastonia, NC, United States of America

* jessamyn.bowling@uncc.edu

**Data Availability Statement:** https://doi.org/10.6084/m9.figshare.19448210.v1.

**Funding:** The authors received no specific funding for this work.

## Abstract

Campus mass shootings have become a pressing policy and public health matter. Twitter is a platform used for processing events among interested community members. Examining the responses of invested community members to a mass shooting on a college campus provides evidence for how this type of violence affects the immediate community and the larger public. These responses may reflect either content (e.g. context-specific) or emotions (e.g. humor). **Aims** Using Twitter data, we analyzed the emotional responses as well as the nature of non-affective short-term reactions, in response to the April 2019 shooting at UNC Charlotte. **Methods** Drawn from a pool of tweets between 4/30/19-5/7/19, we analyzed 16,749 tweets using keywords related to the mass shooting (e.g. "shooting," "gun violence," "UNC Charlotte"). A coding team manually coded the tweets using content and sentiment analyses. **Results** Overall, 7,148 (42.67%) tweets contained negative emotions (e.g. anger, sadness, disgust, anxiety), 5,088 (30.38%) contained positive emotions (e.g. humor, hope, appreciation), 14,892 (88.91%) were communal responses to the shooting (e.g. prayers, healing, victim remembrance), 8,329 (49.73%) were action-oriented (e.g. action taken, policy advocacy), and 15,498 (92.53%) included information (e.g. death/injury, news). All tweets except positive emotions peaked one day following the incident. **Conclusions** Our findings point to peaks in most emotions in the 24 hours following the event, with the exception of positive emotions which peaked one day later. Social media responses to a campus shooting suggest college preparedness for immediate deployment of supportive responses in the case of campus violence is needed.

## Introduction

Mass shootings on college campuses have multiplied in frequency since the 2007 Virginia Tech campus shooting. One survey found that 76 US gun violence incidents happened on college campuses between 2013 and 2016, leading to over 100 casualties [1]. There are direct effects of experiencing campus shootings, including injury, death, and mental health disorders

**Competing interests:** The authors have declared that no competing interests exist.

such as post-traumatic stress disorder and depression [2]. Additionally, campus shootings receive disproportionate media coverage [3], raising concerns about secondary effects on the affected community as well as on contagion effects to others at risk for violence or mental health concerns [4].

As many Twitter posts seek affiliation with others [5], this type of social media can be used for communal responses. Twitter is commonly used during crises and natural disasters to disseminate information, call for assistance, mobilize resources, and process complex emotions [6]. Few studies have harnessed this publicly available data in the aftermath of a mass shooting on a college campus to examine content of tweets. Unfortunately, in April 2019, a former student of the University of North Carolina at Charlotte went to campus and shot and killed two students and injured four others [7]. The present study analyzed Twitter data in the week following the mass shooting for content and affect.

## The role of social media in the aftermath of mass casualty events

Jones et al. [8] reviewed the state of the literature concerning use of social media, and specifically Twitter, for mass casualty events. They noted that natural disasters and disaster communications are commonly the subject of social media data analyses. For example, one study looked at social media use during natural disasters, such as a historic flood in South Carolina, finding that Twitter was widely used to alert others of weather conditions, devastation description, and resource distribution [9]. Brandt et al. [9] provide evidence of Twitter's potential to aid in recovery of natural disasters by providing important and tailored information at different stages of the event to users.

Jones et al. [8] also noted that community violence studies employing Twitter analysis in prior work have focused on traumatic responses related to country-level homicide rates [10] and the aftermath of the Sandy Hook shooting [11]. In their Twitter analysis of three separate mass shooting events, these authors found that general negative emotional content, as well as campus shooting event-related negative emotion, increased following the event [8]. Effects were small across case studies (campus shootings in Isla Vista, CA, Flagstaff, AZ, and Roseburg, OR).

Research on Twitter data may focus on the content of messages, frequency of posts, interactions with posts, and networks of users [12]. In one systematic review, over half (56%) of Twitter studies included content analyses and 15% included sentiment analyses [12]. Given the large volume of data available on Twitter, researchers have often used computational linguistic methods or algorithms to analyze tweets [e.g. 13] with some exceptions using smaller samples [e.g. 14, 15]. However, a reliance on algorithms may preclude nuanced understandings of Twitter content.

## Affective responses to mass shooting

Traumatic sequalae are one of the most well-documented impacts of campus shootings. For instance, following the seminal 2007 Virginia Tech shooting, students on campus during this tragic event demonstrated a variety of notable trauma symptoms including re-experiencing, lack of joy, and heightened physiological reactivity within one year of the event [16]. Closer proximity to the shooting was associated with greater post-traumatic stress symptomology, depression, anxiety, anger, and disconnection in students, faculty, and staff [17–19]; this phenomenon was explained by the occurrence of specific thinking styles—negative cognitions and problematic patterns of rumination–following campus shootings [18, 19]. In addition, a study of 44 shootings found that students exposed to school shooting fatalities had increased and persistent antidepressant use within two years of the shooting event [20].

Some survivors experience post-traumatic growth after campus shootings. This phenomena was associated with emotional proximity to the shooting via deliberate efforts to engage in healthy thinking patterns [19], highlighting the interplay of emotional connection to the event, affect, and cognition. We also know that a relationship exists between traumatic stressors, negative affect, and social media use [21], so negative emotions and cognitive reactions to campus shootings, expressed via Twitter may be predictive of larger mental health concerns [22]. These findings suggest a fuller understanding of people's emotional and cognitive responses to a campus shooting is warranted.

## Gaps in the literature and purpose

Prior studies using Twitter data are limited because they track only a generic category of negative affective tweets due to the inherent restrictions of the computational approach employed; the present study addresses this limitation through a detailed qualitative coding of the affective content of tweets, thereby allowing for a more nuanced understanding of emotional responses to campus mass shootings as portrayed on Twitter. Although we focus on a specific shooting incident, the implications for this study apply to other college contexts as we examine national responses to this shooting.

On April 30, 2019, a former student shot and killed two students and injured four others on University of North Carolina at Charlotte's (UNC Charlotte) campus [7]. The current study examines Twitter responses from the invested national community to that campus shooting with the following goals. First, in light of prior studies highlighting traumatic and negative emotions following mass casualty events, we sought to identify the negative emotional content of messages following the UNC Charlotte mass shooting. Second, given that protective factors (e.g., community connectedness) and positive outcomes (e.g., post-traumatic growth) may result from campus shootings, we sought to understand the presence and nature of positive affect as expressed through Tweets following the UNC Charlotte campus shooting. Finally, Twitter can serve as an informational resource after a mass casualty event as well as a back-channel to hold continued dialogue about an event of interest [23, 24]. Calls for prevention and policy are increasing in response to the rise in campus events. For these reasons, we sought to answer the following research question: 1) What is the nature of non-affective short-term reactions to the UNC Charlotte campus shooting?; 1a) How do the reactions fluctuate in the week following the shooting?.

A priori, we expected that the majority of negative affective reactions expressed on Twitter in the immediate aftermath of a school shooting would portray fear, anxiety or worry. Secondarily, we expected a large portion of tweets to express anger at the perpetrator.

## Methods

An initial sample of 2,246,000 tweets was gathered from the day of the UNC Charlotte shooting on April 30, 2019 until May 7, 2019. Data for this research consisted solely of publicly available tweets and was not considered human subjects research by our university's Institutional Review Board. We used Twitter streaming application program interface (API) to collect tweets [25]. These tweets incorporated the keywords "UNCC", "shooting", "shooter", "mass shooting", "gun", "gun violence", "ninerstrong", "charlottestrong", and "UNC Charlotte". Next, a subset of tweets were reviewed to help ensure the coded tweets were specifically related to the UNC Charlotte shooting and were not related to other national incidents of gun violence. Tweets discussing current events at the time such at #SanDiegoSynagogue and #Teen-Mom as well as related keywords (e.g., "gun girl") were removed. Additionally, tweets were also removed if they utilized the word shoot/shooting but were unrelated to gun violence or

UNC Charlotte (e.g., "I refused to let this video of Chris Evans successfully shooting his shot with a black girl die"). In total, 713,262 tweets were removed. Of the remaining 1,532,738 tweets, we then extracted a random sample of 75,000 from the initial pool. The 75,000 tweets were divided among four coders and evaluated for relevancy to UNC Charlotte shooting, Charlotte, NC, national policies, and other incidents of gun violence. From this, a total of 16,749 tweets fit the criteria in the initial sort.

We conducted directed content analysis of tweets, with categories stemming in part from the existing literature rather than only emerging from the data [26]. Concurrently, we conducted manual emotion analysis, which analyzes emotional classes contained in a text. A preliminary codebook was created based on literature searches and preliminary analyses of the most common themes that appeared during the initial sort. Our emotional classes began with basic emotions in the wheel of emotions (joy, sadness, trust, disgust, fear, anger, surprise, and anticipation) [27, 28] which were then refined based on preliminary analyses of emotions present in our sample (e.g., removal of joy, addition of numbness/tiredness). Each tweet could receive multiple codes, no codes (this did not occur in our sample as non-relevant codes were discarded beforehand). Coders were instructed to contact the team for questionable tweets, which were discussed in weekly meetings during the coding period.

Utilizing the preliminary codebook (see S1 Table), two coders coded a random subsample of tweets (*n* = 500) and included only those that were relevant to Charlotte (*n* = 294). We conducted Fleiss' multirater kappas [29] for codes by the two coders (S2 Table). Overall agreement ranged from adequate-to-good for nine codes (i.e., anger, sadness, humor, appreciation, thoughts/prayers, healing/communal response, victim remembrance/honor, blame/responsibility, and policy advocacy; kappas ranged from .52 to 1.00). Rarer codes had less agreement (e.g., hope, disbelief/shock, need support, uncertainty/confusion; kappas ranged from -0.01 to .38). Codes with fewer occurrences are subject to more extreme kappa values [30]. Therefore, in lieu of relying solely on quantitative inter-rater agreement metrics, we subsequently tabulated the overall percent agreement for each code. Such a strategy is consistent with prior literature under coding conditions using a binary coding scheme and rarely endorsed occurrences of an affirmative content category [31]. The overall percent agreement between the two primary coders from the Charlotte-specific Tweets ranged from 95–100%. After coding the initial sample, the two lead coders met with two additional student coders and the project PI faculty member to identify sample tweets which would help to refine and clarify codebook descriptions, add additional emerging codes to the codebook (e.g. "Shooter" for information related to the shooter), and ensure common team understandings [32]. This team re-coded tweets with initial rater discrepancies; the refined codebook was then used for all subsequent analyses. Each of the student coders coded a random subsample of approximately 3,600 tweets and the PI coded 1,953 tweets. Table 1 shows the emotions coded in the tweets and their corresponding descriptions. It should be noted that tweets may include several categories. For example, the tweet: "*ENOUGH*!!! *If you do not believe we have a problem, then fine, but get out of the way. If you believe more guns are the solution, you are wrong. We have a problem in this country and if you argue with that, you are part of the problem*" was categorized as displaying negative affect and a communal response.

## Results

### Frequency and prevalence of categories tweeted after UNC Charlotte shooting

Of the available 16,749 tweets, 7,148 (42.67%) contained negative emotions, 5,088 (30.38%) contained positive emotions, 14,892 (88.91%) were communal responses to the shooting,

**Table 1. Emotion labels and their descriptions.**

| Emotion Label | Description |
|---|---|
| *Negative Emotions* | |
| Anger/Aggression | Hate; "Your insanity" (name-calling); All caps; Swearing (indicator); "Ridiculous" |
| Numbness/Tired | Descriptions of being done with shootings or responses to shootings. "I don't even know"; "I'm so tired"; "Just another day"; "Too many" |
| Sadness/Grief | Crying emoji; Mourning; "Loss"; Traumatic; Tragedy |
| Disgust/Contempt | "Should"; "Monster"; Moral superiority compared to the perpetrator; Moral judgement related to the situation; "What an embarrassment"; "It's shameful"; Horrific; "Schools should be a safe haven for students and faculty, but again we see a horrific act of violence on a school campus."; "Enough is enough" |
| Anxiety/Worry | Anxious; Concerns about the future or what will come; "Afraid to go back to campus"; "SMH"; "Crazy"; "Senseless" |
| Fear | General fear; NOT related to future |
| Uncertainty/Confusion | Lack of knowing about current situation or future |
| Disbelief/Shock | Surprise at events or state after the events; Can't believe; Unreal |
| *Positive Emotions* | |
| Humor | Jokes about situation or response; Sarcasm[1]; Crying while laughing emoji |
| Hope | Positive ideas about the future; "We can defeat"; "We'll get through this" |
| Appreciation | Appreciation of victims; Gratitude; Hero worship; "He saved multiple lives because of courage"; "Sacrifice" |
| *Communal Response* | |
| Thoughts/Prayers | Blessing; Sending love; "Thoughts and prayers aren't enough"; Prayer hand emojis; Rest in peace/RIP |
| Request/Need Support | Any request/need for support for self or others—any form |
| Healing/Community Response | Responses involving more than individual action; #NinerStrong; vigil; |
| Victim Remembrance/ Honor | Mentions of Riley Howell, Ellis "Reed" Parlier, Drew Pescaro, Sean DeHart, Rami Al-Ramadhan, Emily Houpt |
| Shooter Focus | Information about shooter (Trystan Terrell); about case/arrest |
| Blame/Responsibility | Blame of entities or individuals for gun violence; Not just on people; "Fault" "blame" "on your hands"; "This is what happens when you. . ." |
| *Action* | |
| Action Taken | Individuals/organizations with any action response beyond tweeting to the shooting; Must include some indication that action has been done rather than only suggestion |
| General Action | Generalized calls for something to be done in relation to violence or mass shootings; Enough is enough; end this pandemic; NOT policy specific; NOT individual advocacy; NOT communal response or healing |
| Policy Advocacy | Support for policy change or education; Policies of organizations, campus, national policy/law |
| Individual Advocacy | Advocating for victims or individuals—NOT larger change |
| *Information* | |
| Death/Injury | Reports of deaths/injuries; descriptions of deaths or injuries. |
| | Focus on the body; "Killed"; "loss of life"; "shooting victim"; "dead"; "lose loved ones |
| Official Response | Messages from government, campus (Dean, Chancellor, Campus police), city police |
| Warning Message | Warning others about shooting |
| Reaction to Media | Response to news story or absence of news discussion- NOT social media reactions specifically |
| News | Any news source or article |

[1]Sarcasm usually includes positive affect describing negative situations, and it has been used as a coping mechanism in difficult times to increase well-being [33].

8,329 (49.73%) were action-oriented, and 15,498 (92.53%) included information. Most of the tweets (n = 14,571, 86.99%) were coded with more than one code.

## Categories by day

Fig 1 depicts the total tweets per day by type of category. Note that the shooting occurred at 5:40PM on April 30, 2019. Of the 7,148 negative affect categorized tweets, 4,228 (59.15%) occurred on May 1, 2019. Similarly, of the 14,892 communal response tweets 6,742 (45.27%) were tweeted on May 1, 2019. A majority of the action-oriented tweets ($N$ = 4,188; 50.28%) were posted on May 1, 2019. Information tweets also peaked on May 1, 2019 ($N$ = 6,739, 43.48%). Positive affect tweets peaked a day later, on May 2, 2019: 38.95% ($N$ = 1,982) of all positive tweets occurred then.

Within the peak day for negative affect tweets (May 1, 2019), disgust/contempt was the mostly frequently displayed affect ($N$ = 1,453; 29.50%) followed by numbness (19.75%), anger (16.59%), and then sadness/grief (15.37%). Unexpectedly, fear and anxiety/worry were coded less frequently (less than 5%).

Victim remembrance occurred most often within the communal response category ($N$ = 1,723; 25.56%). Action-oriented tweets were most frequently coded as general action recommendations ($N$ = 1,825; 43.58%) and peaked on May 1, 2019. Within the information category, death/injury ($N$ = 2,400; 35.61%) was most often tweeted. Finally, appreciation was most frequently coded ($N$ = 1,948; 98.28%) within the positive affect category and typically occurred on May 2, 2019. See Table 2 for a complete report and examples of each category coded.

## Discussion

Leveraging social media data (i.e., tweets), we sought to describe the emotional responses as well as the nature of non-affective short-term reactions, in response to the April 2019 shooting

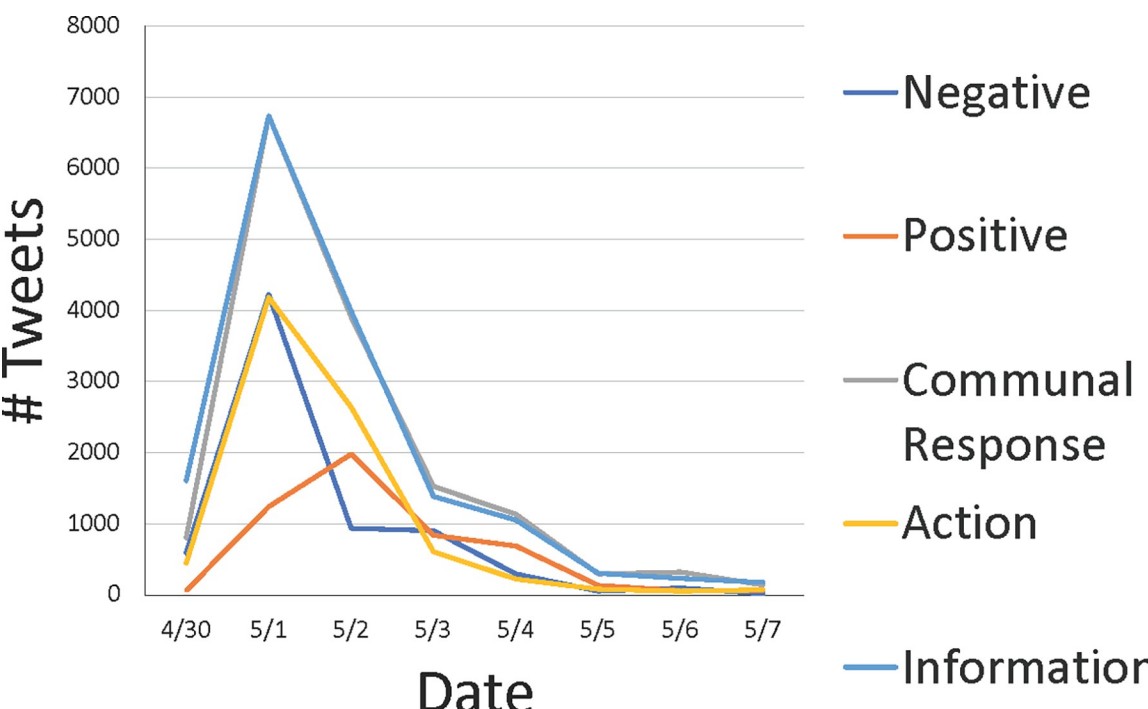

**Fig 1. Total tweets per day by type.**

**Table 2. Peak day frequencies and examples of tweets.**

| Emotion Label | Peak Day* | Examples |
|---|---|---|
| *Negative Emotions* | 5/1/2019 (+1 day) (N = 4,926) | |
| Anger/Aggression | 817 (16.59%) | I know someone who goes to this school. Thank god he and his girlfriend are alright, but this is the SECOND personal connection I have to one of these shootings now (the first one being the movie theater shooting in Aurora, CO). Jesus. Fucking. Christ. |
| Numbness/Tired | 973 (19.75%) | Another school shooting at #UNCC when will it end. . . . |
| Sadness/Grief | 757 (15.37%) | When will #EnoughisEnough actually be enough? Thoughts and prayers are meaningless. Riley died a hero trying to stop the shooter. My heart breaks for his family. #GunControlNow |
| Disgust/Contempt | 1,453 (29.50%) | It is shameful that we are asking our children to choose between running and hiding from and or fighting an active shooter. We should be working to prevent easy access to guns, and protecting our children in the places they should be safest. I am so sorry for the UNCC community. 2 people won't go home tonight after studying all day. they won't take their finals after a semester of hard work. they won't return your texts or calls. they won't have a future because a senseless monster took that from them. Something has to change. #UNCC |
| Anxiety/Worry | 95 (1.93%) | Thoughts and prayers to anyone/everyone affected in the UNCC shooting. . . you never think it could be you. |
| Fear | 133 (2.70%) | Shit. . . I Am scared. . . #UNCC |
| Uncertainty/ Confusion | 189 (3.84%) | This is UNCC last day & on top of that graduation next week. Why tf would anybody wanna do some shit like this smh |
| Disbelief/Shock | 509 (10.33%) | Bro I can't believe there was a shooting on my campus man This picture speaks VOLUMES. All of this still seems soooo unreal. . . I love uncc. . . . this is my home #CharlotteStrong |
| *Positive Emotions* | 5/2/2019 (+2 days) (N = 1,982) | |
| Humor/Sarcasm | 5 (0.25%) | #BREAKING: Well-Regulated Militia Opens Fire At UNCC Charlotte; Cheap Thoughts And Useless Prayers Now Being Rushed To The Scene. . . more on this soon-to-be-forgotten-and-soon-to-be-repeated story as it develops. . . |
| Hope | 29 (1.46%) | UNCC Shooter being taken away, hope everyone is safe |
| Appreciation | 1,948 (98.28%) | Twitter fam,If y'all could send up some prayers for safety for the folks @ UNCC, I'd personally appreciate it. |
| *Communal Response* | 5/1/2019 (+1 day) (N = 6,742) | |
| Thoughts/Prayers | 1,389 (20.60%) | All these school shootings need to stop! Prayers to UNCC |
| Request/Need Support | 602 (8.93%) | @NHLBruins can you please send a shout out to UNC Charlotte student, Drew Pescaro as he recovers from surgery after a gun shot wound? He's a Mass native living in NC and a big B's fan #DrewStrong |
| Healing/Community Response | 901 (13.36%) | Stay safe everyone!! Active shooter at @ucccharlotte campus reported near Kennedy!! #Uncc |
| Victim Remembrance/ Honor | 1,723 (25.56%) | Two people dead from the uncc shooting. Two people who had their whole lives in front of them, maybe even graduation |
| Shooter Focus | 830 (12.31%) | Source tells me: 3 shot at UNCC. Shooter in custody @wsoctv |
| Blame/Responsibility | 1,297 (19.24%) | 3 hurt; assailant in custody. My son is a senior at UNCC—texted me he was safe. Today a university, tomorrow an elementary, middle school or high school. How many lives must be sacrificed for reasonable gun reform to happen? Is there a number the GOP will accept? |
| *Action* | 5/1/2019 (+1 day) (N = 4,188) | |
| Action Taken | 143 (3.41%) | UNCC remains on lock down after shots fired on campus |
| General Action | 1,825 (43.58%) | Can we please get thru one freaking week without a mass shooting?PLEASE??? |
| Policy Advocacy | 1,142 (27.27%) | School shooting at UNC. We need to get rid of this gun free zone. Armed teachers can make a difference |
| Individual Advocacy | 1,078 (25.74%) | 2 students have sat through an entire semester of classes. Taken tests, done homework, studied. . . today is the last day of class. Those 2 students were killed by a senseless shooter, unable to complete their classes, unable to take their finals, unable to have a future. #UNCC |
| *Information* | 5/1/2019 (+1 day) (N = 6,739) | |

*(Continued)*

**Table 2.** (Continued)

| Emotion Label | Peak Day* | Examples |
|---|---|---|
| Death/Injury | 2,400 (35.61%) | BREAKING: At least 6 people taken to hospital after shooting at the University of North Carolina—WCNC-TV |
| Official Response | 1,065 (15.80%) | Please be aware there's a potential active shooter situation at UNCC. Once we have more information, we will share it with you. |
| Warning Message | 52 (0.77%) | NinerAlert: Shots reported near kennedy. Run, Hide, Fight. Secure yourself immediately. Monitor email and |
| Reaction to Media | 2,011 (29.85%) | UNC Shooting:- At least three people shot at University of North Carolina in Charlotte- Suspected Shooter is in c… |
| News | 1,211 (17.97%) | @FOX46News #uncc shooting police near library after evacuation |

*days after event on 4/30/2019

at UNC Charlotte. We did so by employing an intensive qualitative coding process allowing for detailed examination of tweets by subcategory across the days immediately following the campus violence. Two key findings emerged from our analysis. First, we were able to code social media responses into five categories: negative affect, positive affect, communal response, action, and information. Second, all categories but positive affect peaked within one day of the shooting. We consider each set of primary findings in turn.

### Negative versus positive affect in social media responses to a campus shooting

As expected, tweets with negative affect greatly outnumbered tweets with positive affect (humor/sarcasm, appreciation, hope). The timeline of the affect-related communications also differed with tweets with negative affect peaking sharply within 24 hours, followed by positive affect tweets peaking slowly over the second day. This pattern can be interpreted in a variety of ways including being consistent with responses related to the survival instinct (fight/flight/fear first; gratitude and higher order processing second), or the characteristics of this particular social media platform (as a forum for quick dissemination of information and strong [often negative] immediate reactions to on-going events rather than longer term adaptation).

The nature of coded negative tweets was unexpected, however. A priori it was anticipated that a significant percentage of the coded negative affect would include fear, worry, or anxiety as these emotions are typically considered prerequisite experiences for traumatic reactions (DSM-5). However, it is possible that these types of emotions, which may signal immediate and situational vulnerability, may be atypical for Twitter; these feelings may also be hard to express via a tweet during or immediately following the traumatic event. It is also possible that the relatively quick resolution of this shooting event (approximately 15 minutes from shooting to arrest, though campus lockdown lasted much longer) and the time of day in which it occurred (post end of business day) truncated a more robust expression of fear. Understanding the unique affective tone of Twitter, and considering the expression of types of affect in reference to characteristics of a particular trauma will be an important step for future research. It will also be important to determine if those users who expressed fear are more likely to develop negative emotional health sequelae over time.

Contempt was the most common emotional tone conveyed in tweets with negative affect. Contempt is a complex negative emotion that includes the perception that something is beneath consideration or below one's own position. Tweets in this category included at least two distinct types of content: contempt for the shooter and his behavior versus contempt for the university's ability to keep the campus safe or our country's ability to prevent gun violence and protect our children and young adults. Both instances of contempt fit with the

empirically-derived nature of the construct which has been reported to characterized by repudiation and socially excluding the target of the contempt [34]. That is, social media may serve as an outlet for rejection and efforts to exclude the perpetrators of the violence, as well as rejection of the institution's failures to protect its members. This later line of tweets suggests that Twitter provides an outlet for policy-related distress and discouragement as well. Given that Twitter also reaches a large audience, this finding highlights the further need for policy-makers and public health professionals to meaningfully engage social media communities in the aftermath of violent events. The relatively large prevalence of tweets falling into this group suggests a broad awareness of the need for systemic change; this is directly reflected in some tweets that explicitly call out lawmakers to do more to prevent gun violence.

It is also noteworthy that relatively few tweets were categorized as expressing confusion. There seems to be a shift from questioning "why" these events are happening toward expressing, distain and/or distress that policy makers and others have not acted more forcefully to prevent these tragedies. This conclusion is further supported by the substantial number of tweets categorized as angry/aggressive as well as the many positive tweets that were coded as sarcastic humor. Importantly, expressions of aggressive humor (e.g., sarcasm) may actually be a sign of need for help given that such humor styles may be associated with a variety of negative health outcomes [e.g., 35, 36]. Taken as a whole, the affective tweets highlight the diverse nature of the emotional responses occurring in the immediate wake of the UNC Charlotte campus shooting. Moreover, given the peak in negatively valanced Twitter responses in the immediate hours after the event, campus administrators, police, and first responders may need to have prior plans in place to be able to address diverse emotional needs in a time-sensitive manner. The varied responses also support the need for access to well-trained mental health providers in communities affected by gun violence; further research considering how particular early emotional responses relate to long-term adaptation or resiliency is warranted.

## Communal response in social media responses to a campus shooting

The variety of communal response, action and information themes observed in post-shooting Tweets also provide insight into community needs post-shooting. For instance, communal responses highlight the importance of remembering victims, as well as assessing or assigning blame for the shooting and seeking justice. From a public health perspective, to the extent these issues extend beyond the short-term matters, they highlight topics that campuses should be prepared to handle as part of a comprehensive postvention (handling the aftermath of a severe public health event) plan [37]. Although traditionally applied to the topic of suicide [38], postvention planning is necessary for any campus violence-related event. Study findings also inform how campus and community members may be consuming information. That is, information coming from official campus entities was not the most frequently tweeted. Instead, themes of death/injury as well as retweets from off-campus news organizations were more common, perhaps indicating that more people were turning to the news rather than consuming UNC Charlotte official responses. As such, campus communication teams should be working more closely with the local or regional community news outlets for accurate and timely information communication.

Previous research related to mass shootings found local news outlets providing space for grieving and community connection [39]. This may have motivated some of the preponderance of news story sharing, as we saw a variety of news stories from victim remembrance to sharing details of the shooter and other topics. More research is needed to analyze whether the news stories shared were more likely to contain communal grieving spaces or other purposes, such as sharing updates on the facts of the event.

## Time analyses

The fact that most social media response categories trailed off after 24–48 hours is informative. The short-lived nature of social media reactions to the UNC Charlotte campus shooting may reflect a broader shift toward a shortened media cycle. Indeed, it has been hypothesized that the rise in access to 24-hour news and information has resulted in shortened community-level attention paid to even the most intense or tragic events [40]. The short-term nature of social media responses to the UNC Charlotte shooting pale in comparison to the scope of the 2007 Virginia Tech shooting, which at the time drew intense international coverage and long-term media speculation, in part, due to the novelty of the event [41]. No matter the explanation, the short time span of community-level attention afforded such events may point to a missed opportunity for those left behind. As such, one lesson that can be drawn from our findings is the need to prepare immediate response plans to deal with a complex affective and cognitive response to campus shootings that, at first may be expressed through social media but, over time, may recede, resulting in impacted persons falling through the cracks. Formal response plans often, but not always, exist on college campuses in the form of threat assessment teams or post-event protocols involving law enforcement and counseling services [42]. However, they often lack nuance and face a number of implementation challenges (e.g., lack of staff training in postvention; barriers to effective community-wide notification [43]). A more comprehensive approach needs to include best practices such as educating campus community about risk factors/warning signs of violent behavior, making a singular easily identifiable anonymous reporting system available, and a campus-wide team approach involving numerous professionals and departments [44].

## Strengths and limitations

This study incorporated a large pool of tweets and the use of manual coding techniques increasing the reliability of findings. Meetings between the coding team assisted our evolving understandings of concepts and increased team agreement. Due to the limitations in our approach, we may not have captured tweets that did not include our key words or hashtags but were in reference to the UNC Charlotte shooting. Further, in light of the impact of low endorsement of some content codes on formal metrics of inter-rater agreement, we relied in part on overall percent agreement between coders instead. Thus, caution should be applied when interpreting or generalizing findings, especially involving codes with low frequency endorsement. Furthermore, we acknowledge that Twitter users are predominantly younger, wealthier, and more educated compared to the general US population [45]. With our focus on college campuses, the relevant audiences may be inherently more educated. Future research may need to include multiple platforms or communication methods to obtain a more nationally representative portrait.

Limitations to this study should be noted. First, Twitter is a discrete social media platform that tends to be used in particular ways. For example, Twitter is often used for news and high-frequency users often post regarding politics [45–47]. Given the political focus and college students using Twitter may be more socially connected than those using other platforms [48], Twitter may provide early indications of campus climate trends. The volume of tweets suggests that university response needs to include a diverse array of social media platforms. Releasing information on Twitter is likely going to serve a large group that is having a strong, primarily negative, immediate response to campus events.

## Conclusion

We find dense social media posts of negative affect and calling for general action within a relatively short window after a mass shooting. To meet needs implied by this trend, campus

responses for student support must be prepared for immediate deployment. The high proportion of tweets with disgust/contempt, numbness, and anger may reflect the use of Twitter for outward-facing messaging or relate to societal attitude shifts of mass shootings.

## Supporting information

**S1 Table. Preliminary codebook.**
(DOCX)

**S2 Table. Inter-rater reliability with Fleiss multirater kappas.**
(DOCX)

## Author Contributions

**Conceptualization:** Jessamyn Bowling, Erika Montanaro, Robert J. Cramer, Jennifer Langhin-richsen-Rohling, Annelise Mennicke.

**Data curation:** Jessamyn Bowling.

**Formal analysis:** Jessamyn Bowling, Sarai Guerrero Ordonez, Sean McCabe, Shayna Farris, Neielle Saint-Cyr, Wade Glaser.

**Methodology:** Jessamyn Bowling, Sarai Guerrero Ordonez.

**Project administration:** Jessamyn Bowling.

**Supervision:** Jessamyn Bowling, Sarai Guerrero Ordonez.

**Visualization:** Erika Montanaro, Wade Glaser.

**Writing – original draft:** Jessamyn Bowling, Erika Montanaro, Sarai Guerrero Ordonez, Robert J. Cramer, Jennifer Langhinrichsen-Rohling, Annelise Mennicke.

**Writing – review & editing:** Jessamyn Bowling, Erika Montanaro, Sarai Guerrero Ordonez, Robert J. Cramer, Jennifer Langhinrichsen-Rohling, Annelise Mennicke.

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
