## [Decision Letter · Decision Letter 0]

7 Jan 2022

PONE-D-21-34809Coming Together in a Digital Age Community Twitter Responses in the Wake of a Campus ShootingPLOS ONE

Dear Dr. Bowling,

Thank you for submitting your manuscript to PLOS ONE. After careful consideration, we feel that it has merit but does not fully meet PLOS ONE’s publication criteria as it currently stands. Therefore, we invite you to submit a revised version of the manuscript that addresses the points raised during the review process. **Please consider the reviewers' comments below. Additionally, I would like to kindly ask you to consult the data availability policy: https://journals.plos.org/plosone/s/data-availability.**  

We look forward to receiving your revised manuscript.

Kind regards,

Liviu-Adrian Cotfas

Academic Editor

PLOS ONE

Journal Requirements:

2. You indicated that ethical approval was not necessary for your study. Could you please provide further details on why your study is exempt from the need for approval and confirmation from your institutional review board or research ethics committee (e.g., in the form of a letter or email correspondence) that ethics review was not necessary for this study? Please include a copy of the correspondence as an ""Other"" file.

Reviewers' comments:

Reviewer's Responses to Questions

**Comments to the Author**

1. Is the manuscript technically sound, and do the data support the conclusions?

Reviewer #1: Partly

Reviewer #2: Yes

Reviewer #3: Partly

Reviewer #4: Partly

2. Has the statistical analysis been performed appropriately and rigorously? 

Reviewer #1: No

Reviewer #2: Yes

Reviewer #3: N/A

Reviewer #4: No

3. Have the authors made all data underlying the findings in their manuscript fully available?

Reviewer #1: No

Reviewer #2: Yes

Reviewer #3: Yes

Reviewer #4: No

4. Is the manuscript presented in an intelligible fashion and written in standard English?

Reviewer #1: No

Reviewer #2: Yes

Reviewer #3: Yes

Reviewer #4: Yes

5. Review Comments to the Author

Reviewer #1: The paper presents the result of collecting tweets related to

the 2019 UNC Charlotte shooting.

A large number of tweets were first filtered and the resulting subset

then annotated manually by several annotators. As such the paper

describes significant work that can be useful for a variety of purposes

e.g. to do further analysis or for use as a resource for NLP.

However the paper is very vague about many important aspects: what exactly

where the research questions that this works seeks to answer? How exactly

where the annotation guidelines developed and how do they deal with

edge cases, ambiguity, cases not considered in advance?

From the description in the paper it is not clear to me which categories

exactly could get assigned to each tweet (e.g. was it possible to assign

more than one or zero negative emotions to a tweet? Was it possible to

remain undecided? Was at least one label assigned to each tweet for each

of the groups "negative emotions", "positive emotions", "communal response",

"action" etc? The paper mentions in passing that labels from mroe than one

of the groups can be assigned, but there is no precise explanation.

What exactly was the allocation strategy for multiple annotations per

tweet? How many tweets were annotated by each annotator?

There is no proper analysus of inter annotator agreement,

and it is hard to figure out what should have been added given the poor

detail on the annotations and annotation process itself. However it is likely

that nominally scale Krippendorff alpha would probably have been a good

metric to look at. The paper also does not mention how agreement metrics

change e.g. when looking only at presence/absence of negative emotions rather

then detailed negative emotion labels.

It is also not clear how disagreements between annotators were resolved,

the paper states that "consensus codes where then used" but does not

explain how these were derived and what the impact of having disagreement

is on the descriptive statistics derived from them. Nor is there any analysis

of what the variation of the analysis would be based on the disagreements.

Are annotations seen as a manual attempt to find a latend underlying "true

label assignment" or can disagreements be seen as valid different opinions

at least in some cases? What to do about that for the analysis?

The emotion label descriptions in table 1 are rather concise and it is hard to

imagine they would be sufficient for such a complex annotation task, are there

more detailed annotation guidelines and are they part of the data shared?

Unfortunately the data was not at all available for review, access has been prevented

until 2021-12-15. There is also no metadata which would provdide and exact description of

the data, included field and field values that would allow to judge the usefulness of the data.

There is no information e.g. if all the original label assignments by individual annotators

are included and if assignments made by identical annotators can be identified.

It has to be noted that the submission form for this paper states that

"all data are fully available without restriction" however the data is not available at

all at the time of review and the beginning of the paper states "Data will be made available

upon reasonable request to the authors" which could mean anything including that no request

is accepted as reasonable by the authors.

There have to be doubts therefore that the data can be used in any way to replicate the results

in the paper or go beyond them.

The analysis of the manual annotations do not really appear to provide that many new and valuable insights,

when compared to the effort. However, it may be possible to do some further analysis and sharing the

data with the research community in an open and unrestricted way would enable additional

valuable research.

While the paper is written in clear English it is not well structured and often glosses over details

or presents information in a way where it is hard to understand what has been done exactly.

Reviewer #2: A total of 16,749 tweets between 4/30/2019 and 5/7/2019 were manually analyzed to identify the various emotions following the UNC Charlotte mass shooting and to explore the nature of non-affective short-term reactions to the tragic event. This is an overall well-conducted and interesting study. The temporal trend of annotated categories is informative. The authors did a superb job in the Discussion.

Major comments:

1. Can the final annotation guideline/codebook be included as a supplement for review? Some of the categories such as Communal Response appear tricky to distinguish from others. Could any same piece of text be labeled with multiple categories?

2. The separate percentages at the code level (lines 185-187) read a bit confusing in addition to the percentages at the tweet level. How should the code-level relative percentages be interpreted quantitively? As we see there are more negative codes than positive codes, does that mean there was indeed an overall stronger negative sentiment or originally human language has finer semantic granularity for negative expressions? The counting by tweets is easy to follow, but counting by the distinct codes would be affected by whether the denominator makes a meaningful 100%. Please clarify.

Minor comments:

1. Might present distribution of the number of categories per tweet to give a feel about how poly-topic are the contents.

2. Discussion is long and could be organized into subsections.

3. The denominator “available tweets” is unclear in line 217.

4. The relative few worry and fear tweets might also reflect the remote nature of social media. It would be interesting to analyze whether the sentiments varied depending on the distance to the incident (if tractable).

5. Could consider color-coding for Figure 1.

Reviewer #3: The main conclusion of this article is that manual coding of tweets provides a more nuanced understanding of reactions to campus mass shootings compared to computational approaches. However, comparing the current findings based on manual annotations to previous studies using computational methods (Jones et al. 2016; De Choudhury et al. 2014) is not the most convincing. It would have been more persuasive if the authors provided computational analysis of exactly the same tweets and showed readers how manual coding is more advantageous. It is also worth noting that both studies on tweeting negative emotions cited here (i.e. Jones et al. 2016; De Choudhury et al. 2014) were published more than 5 years ago. Given how rapidly machine learning and computational linguistics develop in today’s AI era, it is possible that sentiment analysis could be trained to categorize and analyze linguistic information (and emoji) in a more fine-grained way (e.g. Tian et al. 2020; Lekhtman et al. 2021; Suresh & Ong 2021). It would provide a more significant contribution if the authors show the manual coding is more advantageous to the advanced models of computational sentiment analysis.

It appears that there is no clear cut between some categories. For example, why is the subcategory “blame” a subcategory of “communal response” but not coded as “negative emotions?” Why is “we’ll get through this” coded as “positive emotions” but not under “communal response?” It would be clearer if the authors offer more rigorous criteria for all the categories and subcategories, not simply based on subjective judgments.

Lastly, some examples may fall into more than one category in the coding system. Would that also be possible for one tweet to fall into more than two categories? I am curious about how the overlapping looks like in terms of statistics. Could the overlapping possibly be visualized? The relevant findings would add more layers to the results and deepen the understanding of how different types of non-affective and/or affective reactions would be more likely to occur in the same tweet or a series of reactions after campus violence.

References:

1. Tian, H., Gao, C., Xiao, X., Liu, H., He, B., Wu, H., ... & Wu, F. (2020). SKEP: Sentiment knowledge enhanced pre-training for sentiment analysis. arXiv preprint arXiv:2005.05635

2. Lekhtman, E., Ziser, Y., & Reichart, R. (2021, November). DILBERT: Customized Pre-Training for Domain Adaptation with Category Shift, with an Application to Aspect Extraction. In Proceedings of the 2021 Conference on Empirical Methods in Natural Language Processing (pp. 219-230).

3. Suresh, V., & Ong, D. C. (2021, September). Using Knowledge-Embedded Attention to Augment Pre-trained Language Models for Fine-Grained Emotion Recognition. In 2021 9th International Conference on Affective Computing and Intelligent Interaction (ACII) (pp. 1-8). IEEE.

Reviewer #4: In this article, the authors have attempted to perform a content analysis on tweets posted in response to the UNC Charlotte mass shooting event. The authors employed four coders for the manual classification exercise and identified five broad categories: negative affect, positive affect, communal response, action, and information. The authors claim that the study findings reflect the need for university administrators to immediately allay the negative emotions of students in social media. They claim that manual coding is more suitable than computational analysis for Twitter content analysis.

Although the study has certain right justifications, there are multiple issues that need to be addressed before the paper is of publishable quality. The comments are provided below.

1) The authors claim the data has been acquired from public sources. Specific details need to be provided.

2) The breakdown from the 75,000 random tweets to the final 16,749 tweets need to be explained more clearly. The sentence “Tweets were divided among four coders and

evaluated for relevancy to UNC Charlotte shooting, Charlotte, NC, national policies, and other incidents of gun violence” means that 75000 was split among four coders. But it doesn’t seem to be the case? Did all the four coders work on the same set of tweets or different. I would recommend rewriting this portion

3) In multiple instances across the document, the term “sentiment analysis” has been mentioned. However, sentiment polarity is not used in the methods to identify the sentiment of the tweets. Rather, the authors have used emotion classes. Hence, I would suggest refraining the usage of term “sentiment analysis” in this manuscript

4) In studies where manual coding is performed, reliability sampling based coding is generally performed as a part of preparation. The authors need to mention details about this exercise if it has been performed. If not performed, are there any justifications?

5) Details need to be provided on how the intercoder agreement was improved. I don’t think it is a good practice to justify separate kappa scores for different codes. Please provide one definite overall kappa score for all the codes. The authors are advised to insert a new table for reporting kappa scores as it is a critical part of the manuscript.

6) I am not sure whether the category codes - negative affect, positive affect, communal response, action, and information are mutually exclusive. The authors need to justify how the latter codes have no affect in the tweets.

7) The authors need to cite the sources from which the emotion classes were adopted, since there are multiple emotion class schemes put forth in literature

8) Lines 184-185 in Page 10 – “total of 50,955 category 185 codes were applied to the tweets”. What does this sentence mean?

9) Lines 217-218 in Page 14 – “These categories encapsulated 67.94% (50,955) of available tweets” – What about the remaining tweets?

10) Sarcasm is not always a positive emotion as the authors have attributed it in the results. In many cases, sarcastic tweets have negative connotations and consequences. Authors could think about this phenomenon and make changes accordingly.

11) The implications of this study are not strong in its current form. The authors need to provide more information on how future studies could be benefit from this study.

12) The authors claim manual coding is better suited to Twitter analysis than computational approaches. They cannot make this claim without providing evidence. Perhaps, they could do comparison between manually coded tweets and automatically coded tweets for the different emotion classes.

13) The coded data should be made available for public access.

6. PLOS authors have the option to publish the peer review history of their article (what does this mean?). If published, this will include your full peer review and any attached files.

Reviewer #1: **Yes: **Johann Petrak

Reviewer #2: No

Reviewer #3: No

Reviewer #4: No

---

## [Author Response · Author response to Decision Letter 0]

30 Mar 2022

We have edited the formatting as requested.

2. You indicated that ethical approval was not necessary for your study. Could you please provide further details on why your study is exempt from the need for approval and confirmation from your institutional review board or research ethics committee (e.g., in the form of a letter or email correspondence) that ethics review was not necessary for this study? Please include a copy of the correspondence as an ""Other"" file.

Twitter data are publicly available, and therefore not human subjects research as per our institutional review board. We have uploaded the correspondence as requested. 

Reviewer #1: The paper presents the result of collecting tweets related to

the 2019 UNC Charlotte shooting.

A large number of tweets were first filtered and the resulting subset

then annotated manually by several annotators. As such the paper

describes significant work that can be useful for a variety of purposes

e.g. to do further analysis or for use as a resource for NLP.

We appreciate the reviewer's acknowledging the variety of applications for this work.

However the paper is very vague about many important aspects: what exactly

where the research questions that this works seeks to answer? How exactly

where the annotation guidelines developed and how do they deal with

edge cases, ambiguity, cases not considered in advance?

From the description in the paper it is not clear to me which categories

exactly could get assigned to each tweet (e.g. was it possible to assign

more than one or zero negative emotions to a tweet? Was it possible to

remain undecided? Was at least one label assigned to each tweet for each

of the groups "negative emotions", "positive emotions", "communal response",

"action" etc? The paper mentions in passing that labels from mroe than one

of the groups can be assigned, but there is no precise explanation.

We appreciate the reviewer's request for more detail on our processes. On page 6, line 132 we have added in the following research questions: "For these reasons, we sought to answer the following research question: 1) What is the nature of non-affective short-term reactions to the UNC Charlotte campus shooting?; 1a) How do the reactions fluctuate in the week following the shooting?." We have added more information about the development of the codebook, and how it was augmented after initial analyses. We have added in the following to clarify how coders addressed each tweet: "Each tweet could receive multiple codes, no codes (this did not occur in our sample as non-relevant codes were discarded beforehand). Coders were instructed to contact the team for questionable tweets, which were discussed in weekly meetings during the coding period." (p8)

What exactly was the allocation strategy for multiple annotations per

tweet? How many tweets were annotated by each annotator?

We have added in information about the number in each of the coders' subsamples, which were randomly selected. "Each of the student coders coded a random subsample of approximately 3,600 tweets and the PI coded 1,953 tweets." (p8 L184).

There is no proper analysus of inter annotator agreement,

and it is hard to figure out what should have been added given the poor

detail on the annotations and annotation process itself. However it is likely

that nominally scale Krippendorff alpha would probably have been a good

metric to look at. The paper also does not mention how agreement metrics

change e.g. when looking only at presence/absence of negative emotions rather

then detailed negative emotion labels.

We approached this analysis from a qualitatively driven perspective. Computations of inter-rater agreement are debated among qualitative researchers (e.g. O'Connor & Joffe, 2020; Morse, 2018). Statistical approaches, though many quantitatively-focused researchers rely heavily on them, may have drawbacks (such as how they deal with chance agreement or difficulty in interpretation; O'Connor & Joffe, 2020). Rather than relying on the statistics of the inter-rater reliability, we emphasized our group discussion and understanding which is reflected in the codebook. To improve transparency, we have added our preliminary codebook as Appendix A. 

O’Connor, C., & Joffe, H. (2020). Intercoder reliability in qualitative research: debates and practical guidelines. International journal of qualitative methods, 19, 1609406919899220.

Morse, J. M. (2018). Reframing rigor in qualitative inquiry. In Denzin, N. K., Lincoln, Y. S. (Eds.), The SAGE handbook of qualitative research (pp. 796–817). Thousand Oaks, CA: Sage.

It is also not clear how disagreements between annotators were resolved,

the paper states that "consensus codes where then used" but does not

explain how these were derived and what the impact of having disagreement

is on the descriptive statistics derived from them. Nor is there any analysis

of what the variation of the analysis would be based on the disagreements.

Upon review, we agree with the reviewer that the term "consensus codes" was confusing. We have revised the sentence to read "the refined codebook" instead of "consensus codes". The previous sentences describe how the refined codebook was derived - from group discussion. We follow similar methodologies published using Twitter data and qualitative approaches such as: Jenkins, E. M., Zaher, Z., Tikkanen, S. A., & Ford, J. L. (2019). Creative identity (re) Construction, creative community building, and creative resistance: A qualitative analysis of queer ingroup members' tweets after the Orlando Shooting. Computers in Human Behavior, 101, 14-21.

Are annotations seen as a manual attempt to find a latend underlying "true

label assignment" or can disagreements be seen as valid different opinions

at least in some cases? What to do about that for the analysis?

The reviewer here brings up an important epistemological question. Our team did operate under the premise that disagreements could be valuable in some cases; hence our emphasis on the group discussion rather than statistical methods. We believe that the patterns in the data identified hold true, even accounting for some disagreement. 

The emotion label descriptions in table 1 are rather concise and it is hard to

imagine they would be sufficient for such a complex annotation task, are there

more detailed annotation guidelines and are they part of the data shared?

We have reviewed our notes from our discussion to add further detail to the codebook. The codebook was designed to hold our notes to ensure the team had an understanding of the codes. Our group discussion was a key part of creating this understanding. The codebook was designed for internal team use, rather than to create a tool that could be used by others. The descriptions were sufficient for our team's agreement.

Unfortunately the data was not at all available for review, access has been prevented

until 2021-12-15. There is also no metadata which would provdide and exact description of

the data, included field and field values that would allow to judge the usefulness of the data.

There is no information e.g. if all the original label assignments by individual annotators

are included and if assignments made by identical annotators can be identified.

It has to be noted that the submission form for this paper states that

"all data are fully available without restriction" however the data is not available at

all at the time of review and the beginning of the paper states "Data will be made available

upon reasonable request to the authors" which could mean anything including that no request

is accepted as reasonable by the authors.

There have to be doubts therefore that the data can be used in any way to replicate the results

in the paper or go beyond them.

We have made data available at the link and the restriction is no longer in place. 

The analysis of the manual annotations do not really appear to provide that many new and valuable insights,

when compared to the effort. However, it may be possible to do some further analysis and sharing the

data with the research community in an open and unrestricted way would enable additional

valuable research.

We have reviewed our discussion section to identify places to clarify the insights provided by this study. We have made our data available such that others can conduct further analyses as desired.

While the paper is written in clear English it is not well structured and often glosses over details

or presents information in a way where it is hard to understand what has been done exactly.

We have added additional methodological details to improve the specificity. We have exposed the paper to two more levels of review and changed things such as providing our kappas by code, including details about how coders treated each tweet with options to not code at all, and others. We have added headings in the discussion and reviewed the manuscript for overall flow.

Reviewer #2: A total of 16,749 tweets between 4/30/2019 and 5/7/2019 were manually analyzed to identify the various emotions following the UNC Charlotte mass shooting and to explore the nature of non-affective short-term reactions to the tragic event. This is an overall well-conducted and interesting study. The temporal trend of annotated categories is informative. The authors did a superb job in the Discussion.

We appreciate the reviewer's acknowledgement of the usefulness of the temporal trend analysis and the discussion section.

Major comments:

1. Can the final annotation guideline/codebook be included as a supplement for review? Some of the categories such as Communal Response appear tricky to distinguish from others. Could any same piece of text be labeled with multiple categories?

The final codebook is what was included in the text. We can include the initial preliminary codebook to show our evolution, if that is desired by reviewers (although this would be atypical for qualitative analyses). We have reviewed the codebook and our notes to augment places from our group discussion. Yes, excerpts could be labeled with multiple categories, as we state on page 8 line 184 "It should be noted that tweets may include several categories"

2. The separate percentages at the code level (lines 185-187) read a bit confusing in addition to the percentages at the tweet level. How should the code-level relative percentages be interpreted quantitively? As we see there are more negative codes than positive codes, does that mean there was indeed an overall stronger negative sentiment or originally human language has finer semantic granularity for negative expressions? The counting by tweets is easy to follow, but counting by the distinct codes would be affected by whether the denominator makes a meaningful 100%. Please clarify.

We apologize for the confusion. Our intent including the code level information was to indicate that tweets had more than one code applied to them. We recognize that this was hard to interpret and have since removed this section. 

Minor comments:

1. Might present distribution of the number of categories per tweet to give a feel about how poly-topic are the contents.

We have included the overall percentage of tweets that were coded with more than one code.

2. Discussion is long and could be organized into subsections.

We have added subheadings and shortened the discussion as a whole.

3. The denominator “available tweets” is unclear in line 217.

We have removed this sentence. As the reviewer points out it adds confusion and detracts from the overall point of the paragraph.

4. The relative few worry and fear tweets might also reflect the remote nature of social media. It would be interesting to analyze whether the sentiments varied depending on the distance to the incident (if tractable).

We agree that this is an interesting question. Given the inconsistent nature of geotagging within the tweets, this is outside of the scope of the current paper.

5. Could consider color-coding for Figure 1.

We have added a color-coded Figure.

Reviewer #3: The main conclusion of this article is that manual coding of tweets provides a more nuanced understanding of reactions to campus mass shootings compared to computational approaches. However, comparing the current findings based on manual annotations to previous studies using computational methods (Jones et al. 2016; De Choudhury et al. 2014) is not the most convincing. It would have been more persuasive if the authors provided computational analysis of exactly the same tweets and showed readers how manual coding is more advantageous. It is also worth noting that both studies on tweeting negative emotions cited here (i.e. Jones et al. 2016; De Choudhury et al. 2014) were published more than 5 years ago. Given how rapidly machine learning and computational linguistics develop in today’s AI era, it is possible that sentiment analysis could be trained to categorize and analyze linguistic information (and emoji) in a more fine-grained way (e.g. Tian et al. 2020; Lekhtman et al. 2021; Suresh & Ong 2021). It would provide a more significant contribution if the authors show the manual coding is more advantageous to the advanced models of computational sentiment analysis.

We appreciate the reviewer highlighting this oversight. As other reviewers have mentioned, we agree that the argument about manual versus computational methods was problematic. As such, we have removed it from this paper and pivoted our impact to focus more on responses to mass shootings.

It appears that there is no clear cut between some categories. For example, why is the subcategory “blame” a subcategory of “communal response” but not coded as “negative emotions?” Why is “we’ll get through this” coded as “positive emotions” but not under “communal response?” It would be clearer if the authors offer more rigorous criteria for all the categories and subcategories, not simply based on subjective judgments.

In calling for less of a subjective judgement, the reviewer appears to be preferencing quantitative approaches. Qualitative analyses work toward shared understanding among a team of coders to examine patterns. Our criteria are rigorous in that our team had an understanding of the codebook, even though the excerpts may fall into more than one code.

We agree that there are many nuances to the concepts we're addressing in this work. We have placed blame in communal response due to the inherent social nature of blame and although it includes emotion (such as anger) it is beyond only anger. We based our understanding in part on literature such as: Malle, B. F., Guglielmo, S., & Monroe, A. E. (2014). A theory of blame. Psychological Inquiry, 25(2), 147-186. 

In the example provided by the reviewer, "We'll get through this" would be coded with emotion (which is why it is included where it is in the codebook as an example of hope) as well as healing/community response. 

Lastly, some examples may fall into more than one category in the coding system. Would that also be possible for one tweet to fall into more than two categories? I am curious about how the overlapping looks like in terms of statistics. Could the overlapping possibly be visualized? The relevant findings would add more layers to the results and deepen the understanding of how different types of non-affective and/or affective reactions would be more likely to occur in the same tweet or a series of reactions after campus violence.

We have provided the overall percentage of tweets that were coded with more than one code to help readers understand the nature of the tweets.

References:

1. Tian, H., Gao, C., Xiao, X., Liu, H., He, B., Wu, H., ... & Wu, F. (2020). SKEP: Sentiment knowledge enhanced pre-training for sentiment analysis. arXiv preprint arXiv:2005.05635

2. Lekhtman, E., Ziser, Y., & Reichart, R. (2021, November). DILBERT: Customized Pre-Training for Domain Adaptation with Category Shift, with an Application to Aspect Extraction. In Proceedings of the 2021 Conference on Empirical Methods in Natural Language Processing (pp. 219-230).

3. Suresh, V., & Ong, D. C. (2021, September). Using Knowledge-Embedded Attention to Augment Pre-trained Language Models for Fine-Grained Emotion Recognition. In 2021 9th International Conference on Affective Computing and Intelligent Interaction (ACII) (pp. 1-8). IEEE.

Reviewer #4: In this article, the authors have attempted to perform a content analysis on tweets posted in response to the UNC Charlotte mass shooting event. The authors employed four coders for the manual classification exercise and identified five broad categories: negative affect, positive affect, communal response, action, and information. The authors claim that the study findings reflect the need for university administrators to immediately allay the negative emotions of students in social media. They claim that manual coding is more suitable than computational analysis for Twitter content analysis.

Although the study has certain right justifications, there are multiple issues that need to be addressed before the paper is of publishable quality. The comments are provided below.

1) The authors claim the data has been acquired from public sources. Specific details need to be provided.

Tweets are publicly available and were gathered using an API. We have added a citation for this. We have added in the keywords that were used, such that our approach could be replicated.

2) The breakdown from the 75,000 random tweets to the final 16,749 tweets need to be explained more clearly. The sentence “Tweets were divided among four coders and

evaluated for relevancy to UNC Charlotte shooting, Charlotte, NC, national policies, and other incidents of gun violence” means that 75000 was split among four coders. But it doesn’t seem to be the case? Did all the four coders work on the same set of tweets or different. I would recommend rewriting this portion

We have clarified in this section that the random sample of the 75,000 was divided and evaluated to arrive at the 16,749. The section now reads, "The 75,000 tweets were divided among four coders and evaluated for relevancy to UNC Charlotte shooting, Charlotte, NC, national policies, and other incidents of gun violence. From this, a total of 16,749 tweets fit the criteria in the initial sort."

3) In multiple instances across the document, the term “sentiment analysis” has been mentioned. However, sentiment polarity is not used in the methods to identify the sentiment of the tweets. Rather, the authors have used emotion classes. Hence, I would suggest refraining the usage of term “sentiment analysis” in this manuscript

We appreciate the reviewer's clarification. We have added more detail and removed sentiment analysis. 

4) In studies where manual coding is performed, reliability sampling based coding is generally performed as a part of preparation. The authors need to mention details about this exercise if it has been performed. If not performed, are there any justifications?

We have provided more detail about the type of kappa that was computed and the kappa coefficients have been included in an appendix. As we emphasized a qualitative discussion-oriented approach rather than statistical measures of group understanding, we did not conduct reliability statistics with the full team.

5) Details need to be provided on how the intercoder agreement was improved. I don’t think it is a good practice to justify separate kappa scores for different codes. Please provide one definite overall kappa score for all the codes. The authors are advised to insert a new table for reporting kappa scores as it is a critical part of the manuscript.

As aforementioned, we have provided additional information about our reliability calculations. Kappa scores are often provided for different codes in order to identify areas of disagreement, as the goal is clarity in codes so that the coding team can improve the codebook as well as their own understanding of the codes. We have added the table with the kappa scores as an Appendix (as this is often not included with qualitative manuscripts).

6) I am not sure whether the category codes - negative affect, positive affect, communal response, action, and information are mutually exclusive. The authors need to justify how the latter codes have no affect in the tweets.

In fact they are not mutually exclusive within the excerpts. An excerpt might be both coded within an affect code as well as action and communal response. However, the codes themselves have been described in such a way that they are exclusive from one another. For instance, in "general action" we have included that this is not policy-specific, individual advocacy or communal response. 

7) The authors need to cite the sources from which the emotion classes were adopted, since there are multiple emotion class schemes put forth in literature

We have described our approach with literature that formed our basis for the codes, acknowledging that we then adapted from these emotion classes.

8) Lines 184-185 in Page 10 – “total of 50,955 category 185 codes were applied to the tweets”. What does this sentence mean?

Here we were attempting to describe the number of tweets with multiple codes - we have rephrased this for clarity as: "Most of the tweets (n=14,571, 86.99%) were coded with more than one code."

9) Lines 217-218 in Page 14 – “These categories encapsulated 67.94% (50,955) of available tweets” – What about the remaining tweets?

We have removed this confusing language, as it was attempting to describe the tweets with multiple codes applied to them.

10) Sarcasm is not always a positive emotion as the authors have attributed it in the results. In many cases, sarcastic tweets have negative connotations and consequences. Authors could think about this phenomenon and make changes accordingly.

We agree with the reviewer that sarcasm is hard to categorize - as demonstrated by a wealth of literature focused on natural language processing and sarcasm. We have added a footnote to the table to help explain our use of sarcasm as a positive affect: "Sarcasm usually includes positive affect describing negative situations, and it has been used as a coping mechanism in difficult times to increase well-being (31)". 

11) The implications of this study are not strong in its current form. The authors need to provide more information on how future studies could be benefit from this study.

We have reviewed the discussion for areas in which we can strengthen and clarify the implications of this study. 

12) The authors claim manual coding is better suited to Twitter analysis than computational approaches. They cannot make this claim without providing evidence. Perhaps, they could do comparison between manually coded tweets and automatically coded tweets for the different emotion classes.

We have removed the language focused on manual versus computational approaches. Upon review, we agree that other recent studies using computational approaches also provide nuanced approaches. We have shifted our contribution to focus more on the impacts of mass shootings on college campuses.

13) The coded data should be made available for public access.

As described above, we have made the data available.

---

## [Decision Letter · Decision Letter 1]

7 Sep 2022

PONE-D-21-34809R1Coming Together in a Digital Age Community Twitter Responses in the Wake of a Campus ShootingPLOS ONE

Dear Dr. Bowling,

Thank you for submitting your manuscript to PLOS ONE. After careful consideration, we feel that it has merit but does not fully meet PLOS ONE’s publication criteria as it currently stands. Therefore, we invite you to submit a revised version of the manuscript that addresses the points raised during the review process. Can you please carefully address the concerns raised by Reviewer #5?

We look forward to receiving your revised manuscript.

Kind regards,

Avanti Dey, PhD

Staff Editor

PLOS ONE

Reviewers' comments:

Reviewer's Responses to Questions

**Comments to the Author**

1. If the authors have adequately addressed your comments raised in a previous round of review and you feel that this manuscript is now acceptable for publication, you may indicate that here to bypass the “Comments to the Author” section, enter your conflict of interest statement in the “Confidential to Editor” section, and submit your "Accept" recommendation.

Reviewer #2: All comments have been addressed

Reviewer #5: (No Response)

2. Is the manuscript technically sound, and do the data support the conclusions?

Reviewer #2: Yes

Reviewer #5: No

3. Has the statistical analysis been performed appropriately and rigorously? 

Reviewer #2: Yes

Reviewer #5: No

4. Have the authors made all data underlying the findings in their manuscript fully available?

Reviewer #2: Yes

Reviewer #5: Yes

5. Is the manuscript presented in an intelligible fashion and written in standard English?

Reviewer #2: Yes

Reviewer #5: Yes

6. Review Comments to the Author

Reviewer #2: Minor comments:

1. In the Abstract, the "invested community members" could read ambiguous to some.

2. For Table 2 to be self-contained, a relative distance like +N days after the incident could be noted in addition to the absolute date like 5/1/2019.

Reviewer #5: I should start by noting that I was not part of the original round of reviewers on this manuscript, and so this is my first time seeing it. I did, however, carefully read the reviews/response text included with the review file before reading the paper. On the basis of this, as it turns out I think I can easily see what kinds of things I might have said in a set of comments on the original version, and the fact that other reviewers brought up one of my main points gives me a sense of how the authors would have responded to this.

The biggest issue for me is the lack of rigorous attention to inter-rater reliability in the manual coding, which I think seriously (and to me and many other researchers, I think fatally) compromises the value of the analysis. The authors seem to simultaneously advance an argument that they don't need to attend to inter-rater reliability, while also providing some data on kappas. The discussion of this in the reviewer response memo seems to try to hold up "this is qualitative research" as a kind of cloak that gets them out of providing a lot of useful information on the reliability of the measurements and thousands of individual human decisions upon which the whole paper rests. Connected with this is the assertion that "there is disagreement about reliability metrics," and that there are "drawbacks" to "statistical approaches" to quantifying the uncertainty we have in how reliably the coding principles were applied across thousands of pieces of content. As someone who has conducted a fair amount of manual content analysis research and published pieces using these methods in a variety of outlets across different disciplines and subfields, I honestly found the the whole discussion of this set of issues to be clearly oriented toward a preferred outcome, which is to basically waive off this concern and not have to spend a lot of additional energy revisiting the quality of the data. Yes, O'Connor and Joffe open their piece with the observation that there is disagreement among qualitative researchers on best practices, but they conclude with a recommended workflow that seems different from what the authors used and includes the careful iterated use of "statistical" approaches to carefully calibrating and verifying that the measurement choices made by the coders are reliable enough for us to treat what comes out as signal rather than noise (or at least have a clear sense of the proportion between these). Together, the argument that there is disagreement among qualitative researchers about the value of IRR, and stating that there are "drawbacks" to some methods, strike me as very similar to saying that 'there is disagreement about the causes of climate change,' so we don't have to worry about it.

To be clear, I think there are reasonable and accessible ways to address this set of concerns, and I'd recommend the authors pursue these, rather than just try to waive off the concerns with arguments about how the mere existence of disagreement among some types of researchers means there effectively aren't any rules we need to strictly observe with human coding data, the kappas (a lot of which seem clearly in the 'weak' category by common interpretive schemes for these), and talking about how "rigorous" the discussions of disagreement among some of the coders were. Though I agree with a previous reviewer that Krippendorff is in many ways the superior coefficient/approach, like Kappa it does suffer from a particular drawback that is particularly bad for the kind of data we have here. Specifically, the well-known drawback of measures that seek to correct or account for chance agreement is that they either base expectations for rates of chance agreement on assumptions of random variation (kappa), or they in effect treat the random samples drawn for IRR as a kind of sample of the dataset to produce more informed expectations of what plausible or expected rates of chance agreement might look like (Krippendorff). As a result, when you have presence/absence codes with only two values, and the thing you're looking for the presence of is rare, especially with krippendorff's, it really tanks your reliability estimate because leveraging from these data you end up with a super high estimate of the amount of chance agreement that you need to overcome to show that the coding is reflecting something real in a reliable way. For kappa, you end up applying assumptions about the naturally expected rate of chance agreement that are actually lower than what we might expect based on what we already know about the data (that some coded things are really rare). Generally speaking, unless there is a pretty good sense that the natural occurrence of what you're coding for in the dummy variable is pretty uniformly distributed (i.e. population estimates will land near .5), chance-corrected reliability coefficients are going to come out well below acceptable thresholds a lot of the time, even when there is a high amount of basic agreement between coders, because you only get a few or maybe no chances in any given IRR set to really test whether the two coders can reliably catch that rare occurrence when it appears. There are a couple of ways to deal with this. One is to develop IRR sets that artificially increase the occurrence of the rare thing to bring it closer to a 50/50 distribution in those samples, and then calculate your kappa or alpha (noting of course what you've done and why). This way, you're putting the coders through a chunk of data where each code is ostensibly equally likely, and you will get a much better picture of how well your coding instructions work, which is the point of all of this. Krippendorff himself does not really endorse this though, on the argument that variables with little or no variation are simply of little or no scientific interest. Another way, much more common, is to simply and accurately describe the problem with chance-corrected measures when you have dummy codes and rare events or lopsided distributions, and put forward a combination of simple percent agreement (preferably 90 or 95% or higher is compelling) and a streamlined and more focused discussion of other things you did to deal with this issue. Along with this, you should clearly identify these variables as ones where we should all be cautious in interpreting the results.

While I know I'm a little late in the game here, I would suggest that this last approach for the rare/lopsided variables/codes (note the biases of chance-corrected IRR coefficients with appropriate citations, and provide hopefully very high basic agreement percentages), and then kappa (or Krippendorff's alpha) for cases where the natural distribution of the dummy codes (i.e. closer to 50/50) is a little more friendly to a fair estimate of reliability with a quality, chance-agreement-correcting reliability coefficient is probably the best thing here, in terms of making the findings interesting/useful for the broadest audience and being responsible about the limitations of the data. Absent this, I think we need to take into account everything else that comes along with a strong qual/quant distinction, which is that the data lose a lot of claim to the kind of implications that the authors seem to want to draw out (i.e. that we should pursue specific policies in response to what these data are telling us, because we have high confidence that the results are not dependent on the particular characteristics of the coders and the discussions that they had as they generated the data), and really just offer us large-scale description of what may be going on in these tweets.

Aside from the IRR issue, there are a couple of other areas where I think the manuscript needs substantial additional work.

One is in the limitations of Twitter data for the specific problem as it is framed here. To be sure, the authors note that Twitter is not really representative, though what we know about the skew of Twitter users generally suggests better overlap with "campus community" than for other communities or the public as a whole. I think there is more to this issue that should also be discussed. For example, it's also a reasonable assumption that people on Twitter are more politically interested, and this may be a better explanation for some of the patterns of response. There are substantial literatures on the characteristics of Twitter users and the relationship between Twitter content and media content that would strongly enrich the way that the data are contextualized here.

The second one is that I struggle to see what the data and main take-aways provide that existing literature and common sense do not. The biggest/strongest one seems to be that campuses would do well to anticipate that mass shootings provoke notable spikes in negative emotional states among students and community members right after such events, and that it would be prudent for campuses to have on-demand resources available to deploy quickly in such cases. To my knowledge, most major campuses have something like this in place, and take the chances of such events and the value of pre-planning seriously. If that's incorrect - it would be helpful to include discussion of this.

In closing, I know how difficult it can be to deal with these issues, especially when as much human labor as seems to be the case here have already been put into the dataset. But it's important to realize that effort and attention on their own don't necessarily relate to the quality of the data, and that the kinds of inferences that are being pursued here are important enough that we should have a solid, preferably quantifiable, understanding of the reliability of the measurements on which they rest.

7. PLOS authors have the option to publish the peer review history of their article (what does this mean?). If published, this will include your full peer review and any attached files.

Reviewer #2: No

Reviewer #5: No

---

## [Author Response · Author response to Decision Letter 1]

14 Nov 2022

Comments to the Author

Reviewer #2: Minor comments:

1. In the Abstract, the "invested community members" could read ambiguous to some.

We have changed ‘invested’ to ‘interested’.

2. For Table 2 to be self-contained, a relative distance like +N days after the incident could be noted in addition to the absolute date like 5/1/2019.

We have added this information to Table 2.

Reviewer #5: I should start by noting that I was not part of the original round of reviewers on this manuscript, and so this is my first time seeing it. I did, however, carefully read the reviews/response text included with the review file before reading the paper. On the basis of this, as it turns out I think I can easily see what kinds of things I might have said in a set of comments on the original version, and the fact that other reviewers brought up one of my main points gives me a sense of how the authors would have responded to this.

We appreciate the reviewer’s attention to the previous round of revisions.

The biggest issue for me is the lack of rigorous attention to inter-rater reliability in the manual coding, which I think seriously (and to me and many other researchers, I think fatally) compromises the value of the analysis. The authors seem to simultaneously advance an argument that they don't need to attend to inter-rater reliability, while also providing some data on kappas. The discussion of this in the reviewer response memo seems to try to hold up "this is qualitative research" as a kind of cloak that gets them out of providing a lot of useful information on the reliability of the measurements and thousands of individual human decisions upon which the whole paper rests. Connected with this is the assertion that "there is disagreement about reliability metrics," and that there are "drawbacks" to "statistical approaches" to quantifying the uncertainty we have in how reliably the coding principles were applied across thousands of pieces of content. As someone who has conducted a fair amount of manual content analysis research and published pieces using these methods in a variety of outlets across different disciplines and subfields, I honestly found the the whole discussion of this set of issues to be clearly oriented toward a preferred outcome, which is to basically waive off this concern and not have to spend a lot of additional energy revisiting the quality of the data. Yes, O'Connor and Joffe open their piece with the observation that there is disagreement among qualitative researchers on best practices, but they conclude with a recommended workflow that seems different from what the authors used and includes the careful iterated use of "statistical" approaches to carefully calibrating and verifying that the measurement choices made by the coders are reliable enough for us to treat what comes out as signal rather than noise (or at least have a clear sense of the proportion between these). Together, the argument that there is disagreement among qualitative researchers about the value of IRR, and stating that there are "drawbacks" to some methods, strike me as very similar to saying that 'there is disagreement about the causes of climate change,' so we don't have to worry about it.

To be clear, I think there are reasonable and accessible ways to address this set of concerns, and I'd recommend the authors pursue these, rather than just try to waive off the concerns with arguments about how the mere existence of disagreement among some types of researchers means there effectively aren't any rules we need to strictly observe with human coding data, the kappas (a lot of which seem clearly in the 'weak' category by common interpretive schemes for these), and talking about how "rigorous" the discussions of disagreement among some of the coders were. Though I agree with a previous reviewer that Krippendorff is in many ways the superior coefficient/approach, like Kappa it does suffer from a particular drawback that is particularly bad for the kind of data we have here. Specifically, the well-known drawback of measures that seek to correct or account for chance agreement is that they either base expectations for rates of chance agreement on assumptions of random variation (kappa), or they in effect treat the random samples drawn for IRR as a kind of sample of the dataset to produce more informed expectations of what plausible or expected rates of chance agreement might look like (Krippendorff). As a result, when you have presence/absence codes with only two values, and the thing you're looking for the presence of is rare, especially with krippendorff's, it really tanks your reliability estimate because leveraging from these data you end up with a super high estimate of the amount of chance agreement that you need to overcome to show that the coding is reflecting something real in a reliable way. For kappa, you end up applying assumptions about the naturally expected rate of chance agreement that are actually lower than what we might expect based on what we already know about the data (that some coded things are really rare). Generally speaking, unless there is a pretty good sense that the natural occurrence of what you're coding for in the dummy variable is pretty uniformly distributed (i.e. population estimates will land near .5), chance-corrected reliability coefficients are going to come out well below acceptable thresholds a lot of the time, even when there is a high amount of basic agreement between coders, because you only get a few or maybe no chances in any given IRR set to really test whether the two coders can reliably catch that rare occurrence when it appears. There are a couple of ways to deal with this. One is to develop IRR sets that artificially increase the occurrence of the rare thing to bring it closer to a 50/50 distribution in those samples, and then calculate your kappa or alpha (noting of course what you've done and why). This way, you're putting the coders through a chunk of data where each code is ostensibly equally likely, and you will get a much better picture of how well your coding instructions work, which is the point of all of this. Krippendorff himself does not really endorse this though, on the argument that variables with little or no variation are simply of little or no scientific interest. Another way, much more common, is to simply and accurately describe the problem with chance-corrected measures when you have dummy codes and rare events or lopsided distributions, and put forward a combination of simple percent agreement (preferably 90 or 95% or higher is compelling) and a streamlined and more focused discussion of other things you did to deal with this issue. Along with this, you should clearly identify these variables as ones where we should all be cautious in interpreting the results.

While I know I'm a little late in the game here, I would suggest that this last approach for the rare/lopsided variables/codes (note the biases of chance-corrected IRR coefficients with appropriate citations, and provide hopefully very high basic agreement percentages), and then kappa (or Krippendorff's alpha) for cases where the natural distribution of the dummy codes (i.e. closer to 50/50) is a little more friendly to a fair estimate of reliability with a quality, chance-agreement-correcting reliability coefficient is probably the best thing here, in terms of making the findings interesting/useful for the broadest audience and being responsible about the limitations of the data. Absent this, I think we need to take into account everything else that comes along with a strong qual/quant distinction, which is that the data lose a lot of claim to the kind of implications that the authors seem to want to draw out (i.e. that we should pursue specific policies in response to what these data are telling us, because we have high confidence that the results are not dependent on the particular characteristics of the coders and the discussions that they had as they generated the data), and really just offer us large-scale description of what may be going on in these tweets.

We want to thank the reviewer for this extensive and thoughtful approach to addressing inter-rater reliability. We did calculate simple percent agreement and added this in to our write-up. On page 8, we added the following: “Therefore, in lieu of relying solely on quantitative inter-rater agreement metrics, we subsequently tabulated the overall percent agreement for each code. Such a strategy is consistent with prior literature under coding conditions using a binary coding scheme and rarely endorsed occurrences of an affirmative content category [31]. The overall percent agreement between the two primary coders from the Charlotte-specific Tweets ranged from 95-100%.”

On page 19, we added the following to the limitations: “Further, in light of the impact of low endorsement of some content codes on formal metrics of inter-rater agreement, we relied in part on overall percent agreement between coders instead. Thus, caution should be applied when interpreting or generalizing findings, especially involving codes with low frequency endorsement.”

Aside from the IRR issue, there are a couple of other areas where I think the manuscript needs substantial additional work.

One is in the limitations of Twitter data for the specific problem as it is framed here. To be sure, the authors note that Twitter is not really representative, though what we know about the skew of Twitter users generally suggests better overlap with "campus community" than for other communities or the public as a whole. I think there is more to this issue that should also be discussed. For example, it's also a reasonable assumption that people on Twitter are more politically interested, and this may be a better explanation for some of the patterns of response. There are substantial literatures on the characteristics of Twitter users and the relationship between Twitter content and media content that would strongly enrich the way that the data are contextualized here.

We agree with the political slant, which we had acknowledged previously in the discussion section. As we have focused this paper on how campuses can use Twitter to mount improved responses, we added an additional citation to the sentence: “For example, Twitter is often used for news and high-frequency users often post regarding politics.” We then added this sentence after that: “Given the political focus and college students using Twitter may be more socially connected than those using other platforms, Twitter may provide early indications of campus climate trends”

The second one is that I struggle to see what the data and main take-aways provide that existing literature and common sense do not. The biggest/strongest one seems to be that campuses would do well to anticipate that mass shootings provoke notable spikes in negative emotional states among students and community members right after such events, and that it would be prudent for campuses to have on-demand resources available to deploy quickly in such cases. To my knowledge, most major campuses have something like this in place, and take the chances of such events and the value of pre-planning seriously. If that's incorrect - it would be helpful to include discussion of this.

We appreciate the reviewer’s attention to the take-aways of this paper. We have strengthened this component by added in the following text to the discussion: “Formal response plans often, but not always, exist on college campuses in the form of threat assessment teams or post-event protocols involving law enforcement and counseling services [42]. However, they often lack nuance and face a number of implementation challenges (e.g., lack of staff training in postvention; barriers to effective community-wide notification [43]). A more comprehensive approach needs to include best practices such as educating campus community about risk factors/warning signs of violent behavior, making a singular easily identifiable anonymous reporting system available, and a campus-wide team approach involving numerous professionals and departments [44].”

---

## [Decision Letter · Decision Letter 2]

12 Dec 2022

Coming Together in a Digital Age Community Twitter Responses in the Wake of a Campus Shooting

PONE-D-21-34809R2

Dear Dr. Bowling,

We’re pleased to inform you that your manuscript has been judged scientifically suitable for publication and will be formally accepted for publication once it meets all outstanding technical requirements.

Kind regards,

Nabeel Al-Yateem, PhD

Academic Editor

PLOS ONE

Additional Editor Comments (optional):

Reviewers' comments:

Reviewer's Responses to Questions

**Comments to the Author**

1. If the authors have adequately addressed your comments raised in a previous round of review and you feel that this manuscript is now acceptable for publication, you may indicate that here to bypass the “Comments to the Author” section, enter your conflict of interest statement in the “Confidential to Editor” section, and submit your "Accept" recommendation.

Reviewer #2: All comments have been addressed

Reviewer #5: All comments have been addressed

2. Is the manuscript technically sound, and do the data support the conclusions?

Reviewer #2: (No Response)

Reviewer #5: Yes

3. Has the statistical analysis been performed appropriately and rigorously? 

Reviewer #2: (No Response)

Reviewer #5: Yes

4. Have the authors made all data underlying the findings in their manuscript fully available?

Reviewer #2: (No Response)

Reviewer #5: Yes

5. Is the manuscript presented in an intelligible fashion and written in standard English?

Reviewer #2: (No Response)

Reviewer #5: Yes

6. Review Comments to the Author

Reviewer #2: (No Response)

Reviewer #5: I'd like to commend the authors on a thoughtful revision that I think really addresses the issues I raised in my review.

7. PLOS authors have the option to publish the peer review history of their article (what does this mean?). If published, this will include your full peer review and any attached files.

Reviewer #2: No

Reviewer #5: No

---

## [Editor Report · Acceptance letter]

14 Dec 2022

PONE-D-21-34809R2 

Coming Together in a Digital Age:
Community Twitter Responses in the Wake of a Campus Shooting 

Dear Dr. Bowling:

I'm pleased to inform you that your manuscript has been deemed suitable for publication in PLOS ONE. Congratulations! Your manuscript is now with our production department. 

Kind regards, 

on behalf of

Dr. Nabeel Al-Yateem 

Academic Editor

PLOS ONE